# Pattern generation and symbolic dynamics in a nanocontact vortex oscillator

Myoung-Woo Yoo [1]*, Damien Rontani [2], Jérémy Létang[1], Sébastien Petit-Watelot [3], Thibaut Devolder[1], Marc Sciamanna[2], Karim Bouzehouane [4], Vincent Cros [4] & Joo-Von Kim [1]*

Harnessing chaos or intrinsic nonlinear behaviours of dynamical systems is a promising avenue toward unconventional information processing technologies. In this light, spintronic devices are promising because of the inherent nonlinearity of magnetization dynamics. Here, we demonstrate experimentally the potential for chaos-based schemes using nanocontact vortex oscillators by unveiling and characterizing their waveform patterns and symbolic dynamics using time-resolved electrical measurements. We dissociate nonlinear deterministic patterns from thermal fluctuations and show that the emergence of chaos results in the unpredictable alternation of well-defined patterns. With phase-space reconstruction techniques, we perform symbolic analyses of the time series and show that the oscillator exhibits maximal entropy and complexity at the centre of its incommensurate region. This suggests that such vortex-based systems are promising nanoscale sources of entropy that could be exploited for information processing.

[1] Centre de Nanosciences et de Nanotechnologies, CNRS, Université Paris-Saclay, 10 boulevard Thomas Gobert, 91120 Palaiseau, France. [2] Chaire Photonique, LMOPS EA 4423 Laboratory, Université de Lorraine & CentraleSupélec, 2 rue Edouard Belin, F-57070 Metz, France. [3] Institut Jean Lamour, CNRS, Université de Lorraine, Campus Artem, 2 allée André Guinier, 54011 Nancy, France. [4] Unité Mixte de Physique, CNRS, Thales, Université Paris-Saclay, 1 avenue Augustin Fresnel, 91767 Palaiseau, France. *email: myoung-woo.yoo@c2n.upsaclay.fr; joo-von.kim@c2n.upsaclay.fr

Nonlinear dynamics and chaos are powerful frameworks with which many phenomena in physics, biology, and engineering can be understood[1]. Chaos refers to the high sensitivity of a nonlinear dynamical system to perturbations in its initial conditions, where the temporal evolution is unpredictable on the long term. From the perspective of applications in information processing, chaos has attracted much attention over the last two decades[2] because chaotic waveforms are random-like, yet deterministic and potentially controllable. They have found various applications in information technologies, such as encrypted communications at the physical layer[3], ultrafast random number generation[4,5], data processing[6], computing[7], secure-key exchange[8], radar applications[9], precision sensing[10], and encoding information via symbolic dynamics[11].

Spintronic devices based on magnetic multilayers are good candidates for chaos-based applications, because magnetization dynamics in ferromagnets is intrinsically nonlinear[12–20]. Moreover, such dynamics can be driven and detected by spin-dependent transport phenomena, such as spin-transfer torques, magnetoresistive effects, and (inverse) spin-Hall effects[21,22], giving rise to devices such as spin-torque and spin-Hall nano-oscillators[23,24] that can be integrated into conventional semi-conductor electronics[25]. Because magnetization dynamics can occur at the nanoscale at microwave frequencies, spintronic devices hold much promise for highly compact, GHz-rate information processing using chaos.

One example of chaos in a nanoscale spintronic device can be found in nanocontact vortex oscillators (NCVOs)[26,27]. In the NCVO, the gyration and switching of the vortex core can be induced by spin-transfer torques and oscillating output signals can be detected by the magnetoresistance. In contrast to vortex oscillators based on nanopillars[28–30], the NCVO can exhibit nontrivial dynamics that involves a self-phase-locking phenomenon between the core gyration and core switching[26]. If the ratio between the frequencies of these two processes is irrational, the behaviour is chaotic[27].

Here, we demonstrate experimentally that the chaotic regime of the NCVO involves simple aperiodic waveform patterns. These can be encoded into bit sequences, which are correlated with the core-polarity state of the magnetic vortex. First, we describe time-resolved signals from the NCVO at 77 K and validate their chaotic characteristics from sensitivity to initial conditions and correlation dimension analysis. Then, we show that the time traces are in fact only composed by a few waveform patterns which are ordered aperiodically in the chaotic regime. By reconstructing attractor geometries from the measured time series, we reveal the symbolic dynamics of chaotic NCVOs, which is in good agreement with the patterns observed in simulation. We extract bit sequences based on this symbolic analysis and show that the generated bits can achieve maximal values of the Shannon block entropy and Lempel–Ziv complexity.

## Results

**Chaos in an NCVO.** The NCVO comprises an extended spin-valve multilayer with a metallic point contact (approximately 20 nm in diameter) on the top of the surface (Fig. 1a)[26]. When an electric current is applied through the contact, the component of the current flow perpendicular to the film generates an Oersted field (blue arrow in Fig. 1a) which promotes a magnetic vortex in the free layer and generates a Zeeman energy potential for it that is centred on the nanocontact[31]. The current component in the film plane (orange arrows in Fig. 1a) pushes the vortex core out from the centre by exerting spin-transfer torques[32]. The competition between the two effects results in a stable gyration[26,31,33,34] and switching dynamics[26] of the vortex core around the

nanocontact. In general, the shape of the core trajectory around the nanocontact is not circular[34], as evidenced by a rich harmonic content in the power spectrum. This results from the presence of an antivortex or domain walls in the extended film that appear during the nucleation process[26].

Figure 1b shows a map of the power spectral density of the magnetoresistance oscillations at 77 K as a function of the applied current $I_{dc}$, measured with a spectrum analyser. The NCVO exhibits three dynamical regimes: pure-gyration, commensurate, and incommensurate states[26]. When $I_{dc}$ is lower than a threshold for core reversal ($\sim$10.3 mA here), only the gyration frequency, $f_0$, is observed (red circles in Fig. 1b) with its harmonics ($2f_0, 3f_0, \ldots$) because of the noncircular core trajectories. If $I_{dc}$ is larger than the threshold, core reversal appears in addition to the gyration. This dynamical state is accompanied by additional sidebands at $f_0 \pm f_{mod}$ (yellow crosses in Fig. 1b), where $f_{mod}$ is a modulation frequency that is related to the periodicity of the core reversal. The ratio $f_{mod}/f_0$ as a function of $I_{dc}$ is shown in Fig. 1c, which is distinguished by two plateaus with monotonic increases elsewhere. At the plateaus (yellow regions in Fig. 1c), the frequency spectrum shows clear peaks and $f_{mod}/f_0$ remains constant with $I_{dc}$. In this case, the ratio can be expressed as simple integer fractions (1/3 and 1/4 in this experiment), because the core reversal process is phase-locked to the gyration[26]. The relation between the core dynamics and $f_{mod}/f_0$ is shown in Supplementary Fig. 1 and discussed in more detail in Supplementary Note 1. In the incommensurate state (red regions in Fig. 1c), in contrast, the frequency spectrum becomes more complex, where $f_{mod}/f_0$ varies with $I_{dc}$ from one plateau to another, which indicates that no phase locking occurs between the gyration and core reversal.

The incommensurate state represents chaotic behaviour[27]. Sensitivity to initial conditions, a hallmark of chaos, can be seen in the time traces. These were obtained at 77 K with a single-shot oscilloscope, which were then filtered using a pattern matching technique to reduce the measurement noise (Supplementary Fig. 2, Supplementary Note 2). By overlaying several segments of the time traces with very similar initial conditions, we can obtain a visual measure of the sensitivity to initial conditions in the commensurate (Fig. 1d) and incommensurate states (Fig. 1e). In the commensurate state (Fig. 1d), the waveforms remain coherent over tens of nanoseconds, with evidence of jitter setting in at around 50 ns. In the incommensurate state, however, the coherence is lost below 10 ns (Fig. 1e), which is due to the sensitivity to initial conditions. We can further verify the presence of chaos by analysing the fractal geometry of the reconstructed attractor[35]. To this end, we compute the correlation dimension $D_c$ from the filtered time series using the correlation sum $C_m(\epsilon)$ in Fig. 1f and its derivative with respect to a geometric scaling $\epsilon$ (see Methods). We estimate the geometric dimension in Fig. 1g by looking at constant values of the derivative. For the commensurate case at $I_{dc} = 14$ mA (yellow lines in Fig. 1g) the dimension is found to be $D_c = 1.05 \pm 0.01$, which is very close to 1 and consistent with limit-cycle dynamics of a NCVO presenting a small amount of jitter in the position of the core-polarity switching. For the commensurate case at $I_{dc} = 12.6$ mA (not shown in Fig. 1f–g for clarity), the correlation dimension is $D_c = 1.15 \pm 0.01$, which is further from 1 compared with $I_{dc} = 14$ mA, because of a higher level of jitter in the position of the core-polarization switching. For the incommensurate case (red lines in Fig. 1g), however, the dimension is found to be $D_c = 1.70 \pm 0.04$, which is consistent with a fractal geometry associated with temporal chaos. In Fig. 1h, we present the analysis based on correlation sum to determine a lower bound of the Kolmogorov–Sinai (KS) entropy $h_{KS}$ by computing the $K_2$-entropy metric (see Methods section). In Fig. 1i, by

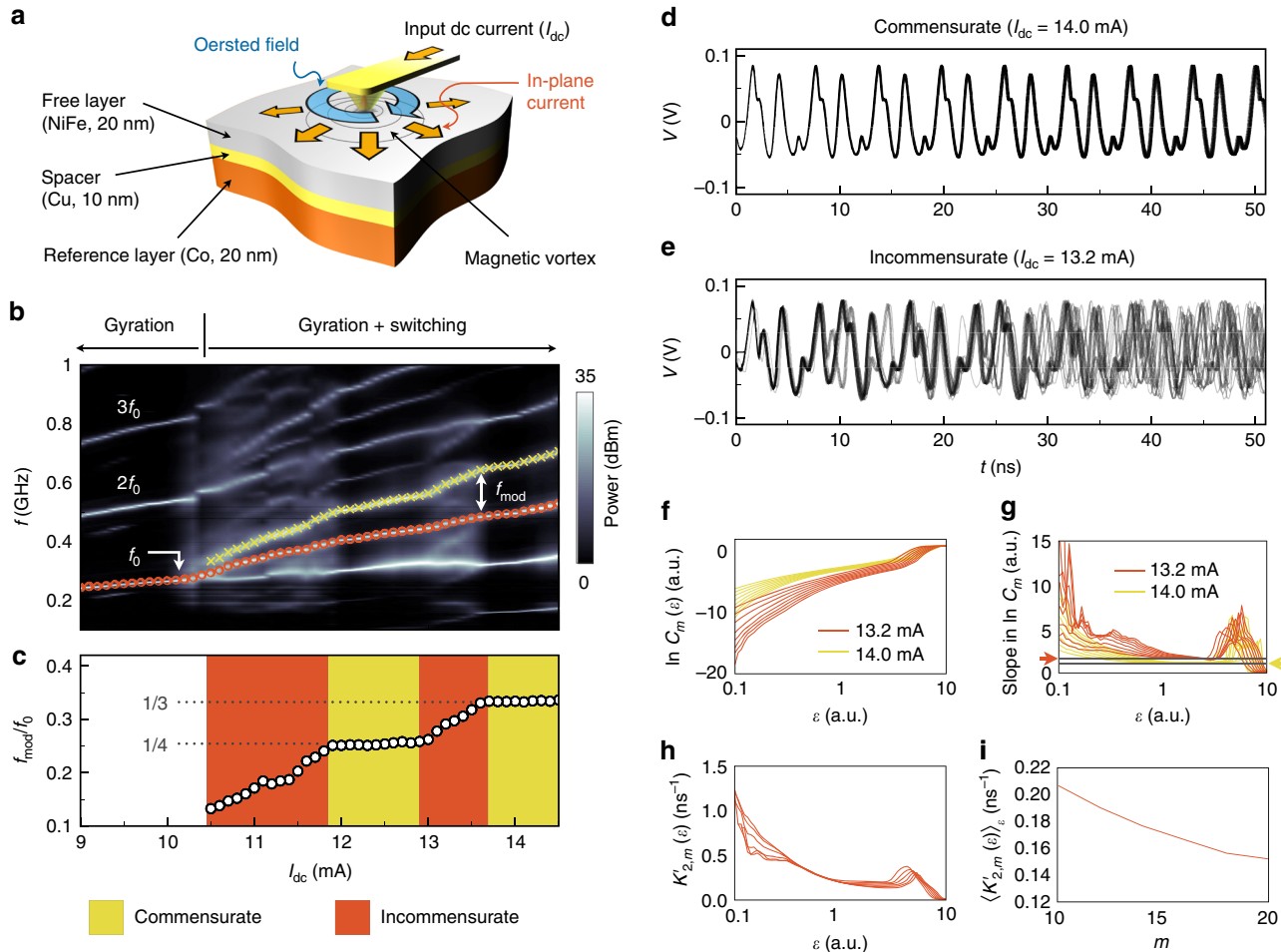

**Fig. 1 Chaotic characteristics of the output time traces. a** A schematic of a nanocontact vortex oscillator. **b** A map of the power spectra as a function of input current amplitudes, $I_{dc}$. The red circles and yellow cross marks indicate a fundamental frequency, $f_0$, and its upper sideband at $f_0 + f_{mod}$, respectively. $f_{mod}$ is a modulation frequency. **c** $f_{mod}/f_0$ as a function of $I_{dc}$. The yellow and red regions represent the commensurate and the incommensurate states, respectively. The dotted horizontal lines indicate plateaus in which the self-phase-locking occurs. **d** Eighteen different time traces which have identical initial conditions in the commensurate state ($I_{dc} = 14.0$ mA). **e** As in **d** but in the incommensurate state at $I_{dc} = 13.2$ mA. **f** Correlation integrals $C_m(\epsilon)$ as a function of a geometric scaling $\epsilon$ and **g** its derivatives of $\partial \ln C_m(\epsilon)/\partial \ln \epsilon$ for embedding dimensions from $m = 10$ (bottom curve) to $m = 24$ (top curve) by increment of 2. The red and yellow curves are for $I_{dc} = 13.2$ and 14.0 mA, respectively. The horizontal lines indicated by red and yellow arrows correspond to estimate of $D_c$ based on the flats in the scaling $\epsilon$-interval $[1, 10^{0.3}]$. **h** Metric $K'_{2,m}(\epsilon)$ as a function of $\epsilon$ for embedding dimensions from $m = 10$ (top curve) to $m = 24$ (bottom curve) by increment of 2 at $I_{dc} = 13.2$ mA and **i** $\langle K'_{2,m}(\epsilon)\rangle_\epsilon$ in the scaling range $[1, 10^{0.3}]$ for the asymptotic determination of the $K_2$-entropy.

extrapolating the tendency of $\langle K'_{2,m}(\epsilon)\rangle_\epsilon$ with increasing values of $m$ with an exponential fit, we estimate that $K_2 = 0.12 \pm 0.02$ ns$^{-1}$ for $I_{dc} = 13.2$ mA. These results are consistent with a previous study using the titration of chaos with added noise[36] to identify the presence of chaos in the NCVO[27]. Note that the responses in the incommensurate state can be reproduced by micromagnetic simulations even at 0 K as shown previously[26]. We contend therefore that the measured chaotic characteristics are mostly deterministic, rather than stochastic as driven by thermal fluctuations.

**Pattern generation**. A feature of the chaos generated by the NCVO involves distinct waveforms that repeat aperiodically. Representative experimental time series are shown in Fig. 2a–c. Here we use a time axis that is normalized with respect to the core gyration period, $1/f_0$, such that $tf_0$ represents the number of core gyrations. Both commensurate and incommensurate states show similar features where distinct oscillatory patterns are delimited by cusps. These patterns correspond to a number

of orbits of the vortex core around the nanocontact, with the cusps representing a core reversal event; the position of these cusps are indicated by the dots and dotted lines in Fig. 2a–c. Note that core reversal results in the change in the sense of gyration (i.e. clockwise to counterclockwise, and vice versa). These features in the measured time series are reproduced in micromagnetic simulations (see Supplementary Fig. 3 and Supplementary Note 3).

From the intervals between the core switching events in Fig. 2a–c, we can define a gyration number for the core switching, $n = \Delta t f_0$, which is shown in Fig. 2d–f. In the commensurate state, $n$ exhibits a simple time evolution. At $I_{dc} = 12.6$ mA, the switching always occurs every two gyrations (Fig. 2d), so $n$ remains constant at 2. Similarly, at $I_{dc} = 14.0$ mA, core reversal occurs after one and two gyrations successively, a process which repeats periodically; in this case $n$ oscillates between 1 and 2 as shown in Fig. 2f. In the incommensurate state ($I_{dc} = 13.2$ mA), however, $n$ switches between 1 and 2 in an aperiodic fashion (Fig. 2e), which is consistent with the chaotic dynamics expected in this regime.

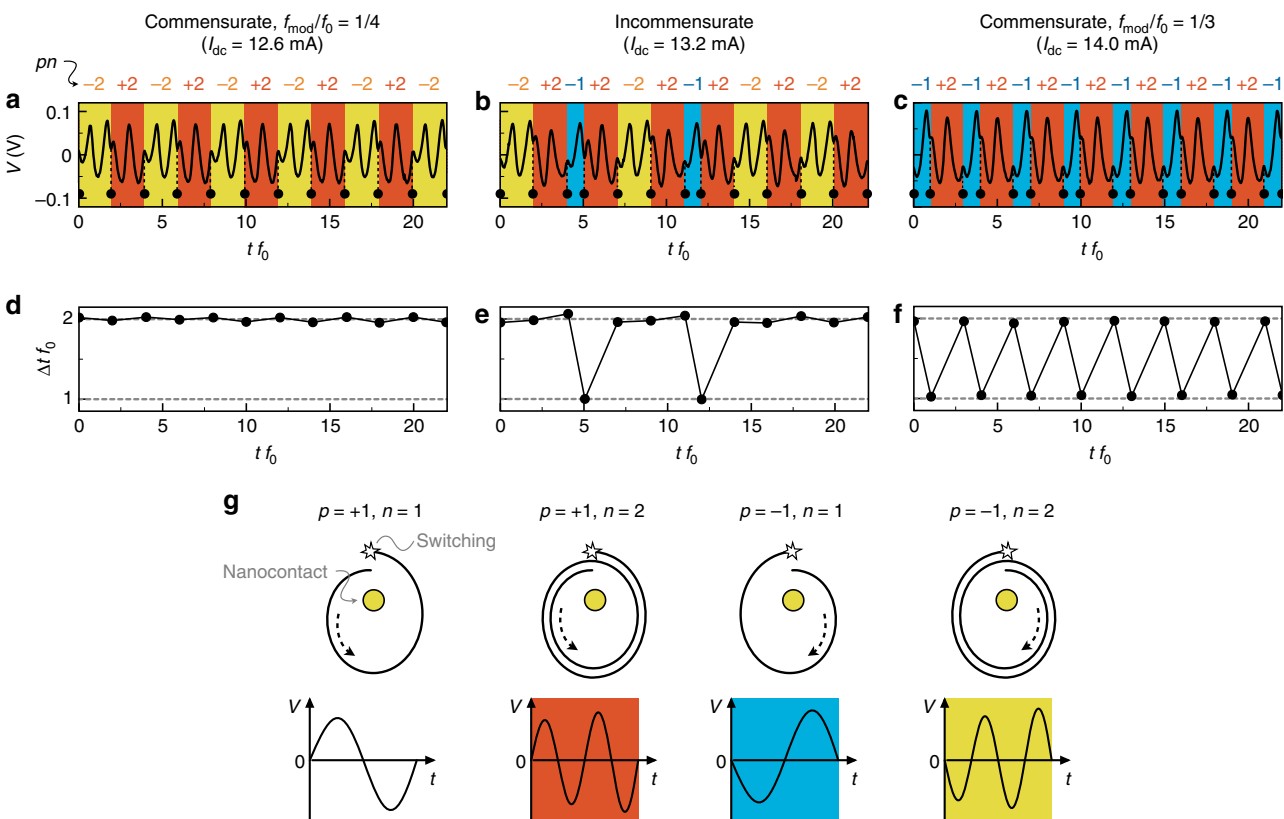

**Fig. 2 Pattern generation from time series. a–c** Representatives of experimentally measured time series at $I_{dc} = 12.6, 13.2,$ and 14.0 mA. The black dots with dotted lines indicate the core-polarity switching events. We normalized the time axes $tf_0$ which is identical with a number of core gyrations. The yellow, blue, and red regions indicate waveform patterns denoted by $pn = -2, -1,$ and $+2$, respectively, where $p$ is a core polarity and $n$ is a required gyration number for the core-polarity switching. **d–f** Time evolutions of a required gyration number for the core switching, $n = \Delta t f_0$, obtained from **a–c** by calculating intervals between the black dots. **g** Schematics of the possible core-polarity switching scenarios. The core trajectories (top panels) and expected output waveforms (bottom panels) are shown for different $p$ and $n$ combinations. The colours of the oscillatory patterns (red, blue, and yellow) correspond with those in **a–c**.

The required gyration number for core switching, $n$, is always approximately integer (typically 1 or 2 in this experiment) in both commensurate and incommensurate cases as shown in Fig. 2d–f. This is consistent with simulation results, in which core-polarity switching occurs only in a restricted region of the film plane close to the nanocontact, where conditions for core reversal are met[26]. In addition the core polarity, $p$, can only have two values, $+1$ and $-1$, so we hypothesize that in general there are only four possible patterns for the commensurate or incommensurate states, $pn = +1, +2, -1,$ and $-2$. In other words, the nonlinear physical properties of the NCVO result in a sequence that represents a combination of these four patterns.

We plot schematic core trajectories of the possible switching scenarios in Fig. 2g, along with the expected time series for different $pn$. Without loss of generality, we assume here that the vortex has clockwise chirality and the reference layer is saturated in the $+y$ direction. Based on the schematic waveforms, we can identify the corresponding oscillatory patterns from the time series, as indicated by the background colours in Fig. 2a–c. In the commensurate state (Fig. 2a, c), the time series are composed of two $pn$ patterns, which repeat periodically. They involve $pn = \{-2, +2\}$ for $I_{dc} = 12.6$ mA (Fig. 2a) and $\{-1, +2\}$ for 14.0 mA (Fig. 2c), respectively. In the incommensurate state (Fig. 2b), there exist three $pn$ patterns, $pn = \{-2, -1, +2\}$, which appear without a well-defined periodicity. This shows

that the NCVO generates simple oscillatory patterns even in the chaotic state.

**Symbolic dynamics and bit sequences**. We further analyse the pattern generation from the perspective of symbolic dynamics[37]. The principle of symbolic dynamics is to find an adequate partition of the system's Poincaré section in its phase space (see Methods), such that every time there is a transition from one region of the section to another, a symbol is emitted. As a result, the nonlinear dynamics of the system can be reduced to a sequence of symbols. However, generating symbols with the proper partitions and surfaces from model-free, experimental, and scalar time series is a challenging problem in general. First, we reconstruct attractor geometries from the measured time series using a three-dimensional delay embedding, which is sufficiently large to completely unfold the reconstructed attractors when considering the commensurate and incommensurate states shown in Fig. 3a–c[35]. In the commensurate state, the NCVO exhibits a limit cycle, because the trace is periodic (Fig. 3a, c). On the contrary, the attractor for the incommensurate state is more intricate (Fig. 3b). As explained in the Methods section, we set a proper Poincaré surface (white surfaces in Fig. 3a–c) allowing for a potential generating partition. Note that the surface is not unique, and we can choose any different plane for the symbolic analysis. On the surface, we obtain the Poincaré maps from the

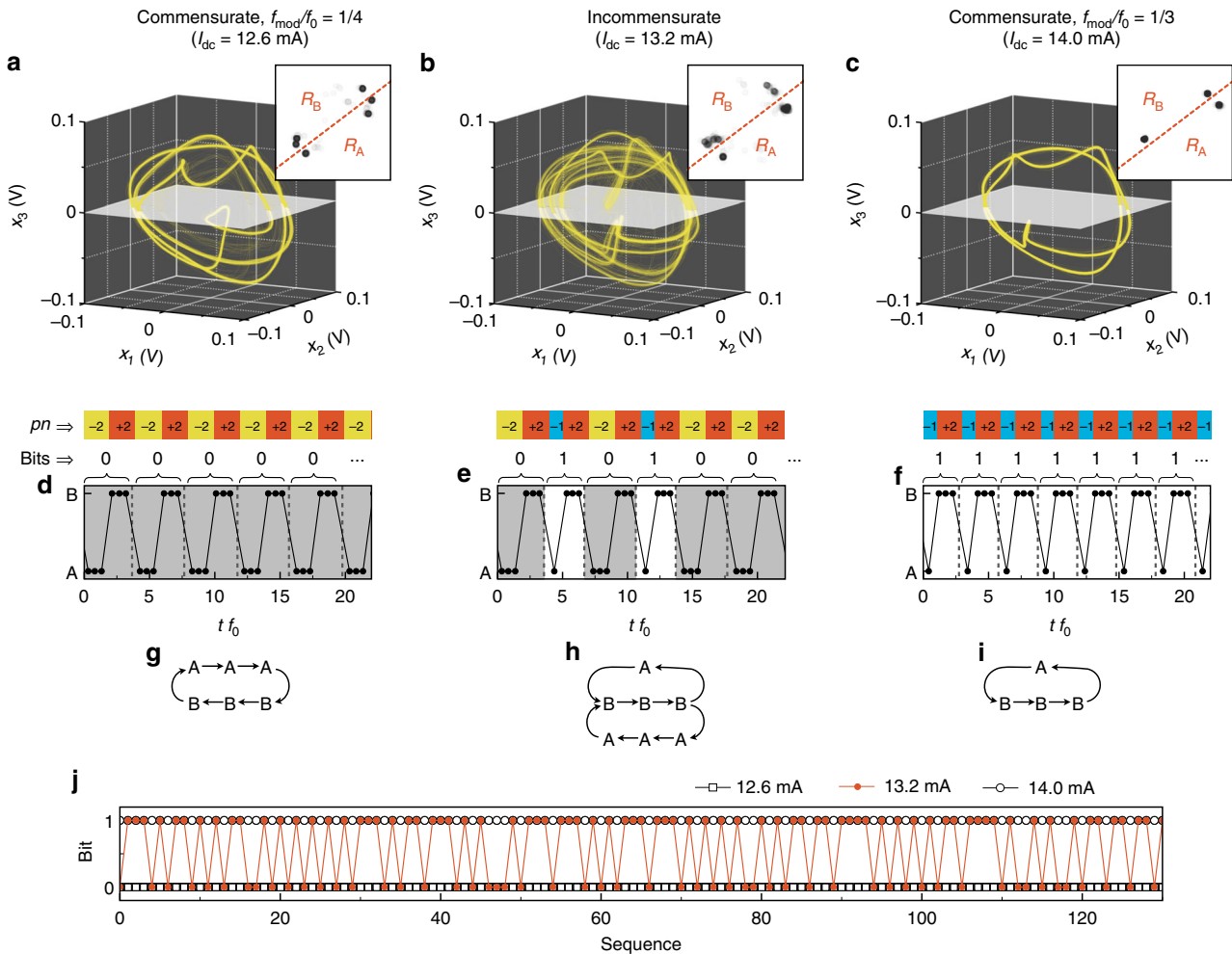

**Fig. 3 Reconstructed attractor geometries and symbolic dynamics. a–c** Reconstructed attractor geometries by a method of delay from the measured time traces at $I_{dc}$ = 12.6, 13.2, and 14.0 mA, respectively. The white plane is arbitrarily chosen Poincaré surface of section (see Methods). (Inset) Poincaré maps at the surfaces. The red dashed lines indicate a simple partition to divide the plane into two regions, $R_A$ and $R_B$, for encoding symbols, A and B. The Poincaré surface of section and partitions are identical for all $I_{dc}$ in these figures. **d–f** Dynamics of symbols defined from the partition on the Poincaré' maps. Above the graphs, corresponding $pn$ patterns and generated bit sequences are represented. The bits are defined as 0 ≡ [A,A,A,B,B,B] and 1 ≡ [A,B,B,B]. **g–i** Rules of the symbolic dynamics at $I_{dc}$ = 12.6, 13.2, and 14.0 mA. **j** Generated bit sequences for long term at $I_{dc}$ = 12.6, 13.2, and 14.0 mA.

intersection (insets of Fig. 3a–c) in which we find several clusters of the points. Note that here, for the Poincaré map, we do not consider the transverse orientation. We use a simple partition to divide the map into two different regions, $R_A$ and $R_B$ (red dashed lines in the insets of Fig. 3a–c), then record the symbols, A or B, when the attractor cross the surface either in $R_A$ or $R_B$, respectively. The encoded symbolic sequences are shown in Fig. 3d–f, where the corresponding $pn$ patterns are shown above the graphs. By comparing the $pn$ patterns and symbolic dynamics, we can see that $pn$ = −2, −1, and +2 correspond to [A,A,A], [A], and [B,B, B], respectively. This result shows that the choice of the partition for the determination of the symbolic dynamics is in good agreement with the $pn$ sequences in both the commensurate and incommensurate states. Other partition choices are also possible but may render identifying the symbolic sequences more difficult (see Supplementary Fig. 4 and Supplementary Note 4).

We can find simple rules in the symbolic sequences (Fig. 3d–f). In the commensurate case, the sequences show only one repeated cycle: [A,A,A,B,B,B] and [A, B, B, B] for $I_{dc}$ = 12.6 and 14.0 mA, respectively (Fig. 3g, i). In the incommensurate state, however, two possible cycles coexist in the sequence and appears erratically over

time (Fig. 3h). To simplify the analysis of complexity, we define binary symbols attributed to the two different patterns accessible by the NCVO: 0 ≡ [A,A,A,B,B,B] and 1 ≡ [A,B,B,B]. Then, we extract the bit sequences as represented in Fig. 2d–f above the graphs. In the commensurate states (Fig. 2d, f), the NCVO generates only one type of bits: 0 for $I_{dc}$ = 12.6 mA and 1 for $I_{dc}$ = 14.0 mA. On the other hand, in the incommensurate state (Fig. 2e), the NCVO generates bits in no apparent order. To better illustrate this, we plot a bit sequence generated in the chaotic regime for a longer duration (Fig. 3j). In the NCVO system, the bits are generated at an average rate of ∼ 131 Mbit s$^{-1}$, which is much faster than random number generation using spintronic devices such as stochastic magnetic tunnel junctions[38,39]. The bits are produced by chaotic polarity switching of the core after a few revolutions around the nanocontact, so the generation rate is mainly proportional to the gyration frequency, $f_0$, which is in the range of 0.1–1 GHz in typical soft magnetic materials. Here, the gyration frequency can reach up to ∼ 600 MHz, so the potential bit generation could reach up to ∼ 300 Mbit s$^{-1}$.

Interestingly, the possible cycles in the incommensurate state correspond to those the NCVO already exhibits in its

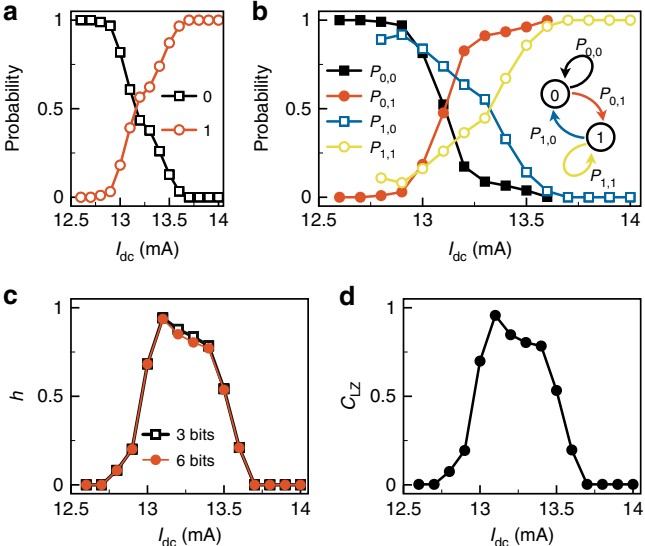

**Fig. 4 Complexity and control of bit sequences. a** Probability of 0 and 1 in the generated bit sequences as a function of $I_{dc}$. **b** Probability of moving from the current state, $i$, to the next state, $j$, $P_{i,j}$. The inset shows the Markov chain for a 1 bit information. **c** Shannon block entropy rate, $h$, as a function of $I_{dc}$ for different block lengths, 3 bits and 6 bits. **d** Normalized Lempel–Ziv complexity, $C_{LZ}$, as a function of $I_{dc}$.

commensurate state at $I_{dc} = 12.6$ and $14.0$ mA, respectively (Fig. 3h). This shows that not only the complexity is driven by the core switching but also that the type of patterns generated in the incommensurate (chaotic) state is fundamentally restricted to accessible patterns associated to the two neighbouring commensurate states. Hence, within the incommensurate region, we anticipate that the probability of appearance of one among the two accessible patterns can be controlled by $I_{dc}$.

**Complexity of the generated bit sequences**. To assess the complexity of the bit sequence extracted from the symbolic dynamics, we compute the probability of each bit as a function of $I_{dc}$ (Fig. 4a). We also estimate the probability of transition from one bit to the next, $P_{i,j}$, where $i$ and $j$ are the current and the next binary states (Fig. 4b), while assuming a Markovian property, i.e., a one-bit memory depth associated to the bit sequence (see Supplementary Fig. 5 and Supplementary Note 5). In Fig. 4a, as $I_{dc}$ increases and the NCVO transitions from a commensurate to an incommensurate state, we observe that the probability of symbol 0 (resp. symbol 1) decreases smoothly from $P_0 = 1$ (resp. increases from $P_1 = 0$) monotonically and reaches the point where $P_0 = P_1 = 0.5$ at about $I_{dc} \approx 13.1$ mA before continuing to decrease to $P_0 = 0$ (resp. to increase to $P_1 = 1$). Similarly, in Fig. 4b, we observe similar behaviour for the transition probabilities and specifically at $I_{dc} \approx 13.1$ mA, we have $P_{0,0} = P_{0,1} \approx 0.5$, $P_{1,0} \approx 0.75$, and $P_{1,1} \approx 0.25$. With this particular transition probability matrix, the entropy rate of the Markov chain describing the NCVO's pattern dynamics can be computed and reads $h_\infty \approx 0.93$ bit per binary symbol for a maximum entropy rate of 1 bit per binary symbol (see Supplementary Note 6). These results imply that for certain DC currents it becomes difficult to predict the next generated bit from the current one in the incommensurate state. But as the current is varied from this operating point, the entropy rate decreases because the NCVO dynamics approaches one of the two neighbouring commensurate states. As such, one of the two accessible

temporal patterns will start to dominate the other and its corresponding bit (0 or 1) will become more probable.

To further assess the unpredictability of the generated bits, we evaluate the complexity of the bit sequences generated by using two metrics from information theory: the Shannon block entropy rate, $h$, and the normalized Lempel–Ziv complexity, $C_{LZ}$. The Shannon block entropy rate measures the amount of uncertainty carried on average by each bit: for a binary source of information, as the one generated by a chaotic NCVO, we have maximum entropy $h_{max} = 1$ bit per binary symbol. We compute the entropy rate as a function of $I_{dc}$ (Fig. 4c) and consider binary blocks of length $n = 3$ and $n = 6$ to obtain more robust estimates, while ensuring the estimation remains invariant with respect to these choices of block lengths (see Methods). We observe that the uncertainty from the bit stream generated by the NCVO is non-monotonic inside the incommensurate region; it gradually increases to reach a peak value of $h \approx 0.94$ at $I_{dc} = 13.1$ mA before decreasing again as the 1/3 commensurate state is approached. This is consistent with an asymmetric distribution for the probability mass function of the generated bit, which indicates that the bit sequence generated by the NCVO in its incommensurate state inherits complexity from the Kolmogorov–Sinai entropy created by the chaotic dynamics[35]. The determination of the lower bound of KS entropy, $h_{KS} \approx 0.12$ ns$^{-1}$ at $I_{dc} = 13.2$ mA, provides the maximum level of entropy extraction from the NCVO at $\sim 100$ Mbit s$^{-1}$: This is consistent with the average generation rate of bits achieved by the symbolic encoding used in our study. We perform a similar analysis for the Lempel–Ziv complexity $C_{LZ}$, which measures the diversity (i.e., lack of redundancy) of binary patterns encountered in a binary sequence. We observe a similar trend with finite values in the incommensurate region, reaching almost the maximum value of $C_{LZ,max} = 1$, while $C_{LZ} \approx 0$ in the commensurate state. This means the bits generated by the chaotic NCVO cannot be efficiently compressed because of its maximal complexity. Since $h$ and $C_{LZ}$ almost attain their maximum values of $\sim 1$, the raw generated bit sequences have suitable statistical features to be considered as a physical source of entropy for information processing. The probability and complexity assessments have been performed on more than 9300 bit strings for each $I_{dc}$, which are obtained by the pattern recognition method shown in Supplementary Fig. 6 and described in Supplementary Note 7.

## Discussion

Based on our analysis, the chaotic magnetization dynamics of the NCVO has desirable properties for information processing applications. First, the complexity of the magnetization dynamics is not encoded in the amplitude of the waveform but in the alternation of regular patterns. In that sense, it natively exhibits very similar properties to the chaos generated by specifically engineered systems proposed in ref. [40] and hence has high resilience to perturbations. The extraction of the bits, once the two patterns are identified and stored digitally, could be done in real time with a low computational cost using matched filters[41] implemented on field programmable-gate arrays or digital signal processors. A recent application of chaotic dynamics similar to those of NCVOs was proposed for WIFI communications (i.e., for challenging environments with multi-path interference, jamming, distortion) using electronic circuits[42,43], because of the capacity to recover more easily regular patterns in low signal-to-noise conditions compared with chaotic waveforms in amplitude and/or phase of the signal. Second, the NCVO generates directly chaotic bits with high level of entropy (e.g., entropy rate of $h_\infty \approx 0.93$ bit per binary symbol at $I_{dc} = 13.1$ mA in the incommensurate state), hence making it a promising technology

for physical random number generation with minimal post-processing. Furthermore, with chaotic effects taking place at the nanoscale, dense integration of NCVOs on a single spintronic chip would allow for the parallel generation of random bits with aggregate rate in the tens or hundreds of Gbit s$^{-1}$ similar to the approach already used in ref. [44] with microelectronic circuits. Finally, the properties of the chaotic NCVO dynamics can be easily tailored with the DC current injected in the nanocontact. As a result, one could design advanced control strategies by injected small perturbation in the current to tune the chaos and encode data in pattern alternation, as described in refs. [11,45], and to create a robust chaos-based encryption at the physical layer.

In addition, knowing the underlying structure of the temporal chaotic dynamics and its connection to the timing of the core-switching events, one could also design experimental strategies to control electrically the core dynamics to encode information secretly. This paves the way for nanoscale chaos-based information processing using the nonlinear dynamics of spintronic devices.

## Methods

**Sample fabrication and measurements**. The nanocontacts are fabricated using the atomic force microscope nano-indentation method[46] on the top of the sputtered deposited multilayer with the composition SiO$_2$/Cu (40 nm)/Co (20 nm)/Cu (10 nm)/Ni$_{81}$Fe$_{19}$ (20 nm)/Au (6 nm)/photoresist (50 nm)/Au (nanocontact)[27]. The diameter of the contact is $\sim 20$ nm. The vortex is first nucleated by reversing the free layer magnetization with an in-plane applied magnetic field in the presence of a $I_{dc} = 16$ mA current applied through the nanocontact. The vortex gyration around the nanocontact results in magnetoresistance oscillations that are detected after amplification as voltage fluctuations in the frequency domain by a spectrum analyser and in the time domain by a single-shot oscilloscope. RF switches are used to connect either of these two equipments to the sample, hence allowing for both time- and frequency-domain measurements to be made sequentially. The experiments are conducted in a cryostat at liquid nitrogen temperature (77 K) to better isolate the chaotic dynamics, which is a deterministic process but can appear as a thermal noise, from thermal fluctuations which are true stochastic processes. Note that all the data in this study are measured using a single device. Further details of the experimental setup and measurement procedure are described elsewhere[26,27].

**Thermal noise filtering from time traces**. To improve the signal-to-noise ratio of the experimental time traces, we used an averaging filter. We collected similar short-term waveforms ($\sim 7.5$ ns) from full time series by calculating convolutions, then averaged over them. This method is applicable in our system because the output time traces are composed of only two or three patterns even in the chaotic regime. More details can be found in Supplementary Fig. 2 and Supplementary Note 2.

**Time-delay embedding and phase-space reconstruction**. A time-delay embedding procedure is used to form an $m$-dimensional vector space related to the original phase space by a diffeomorphism preserving topological invariants of the original attractor, if $m > 2d_A$ with $d_A$ dimension of the original attractor[47]. The vectors in the reconstructed phase space are obtained from univariate time-resolved series as follows:

$$\mathbf{v}_n^{(m)} = [V(t_n), V(t_n - \tau), \dots, V(t_n - (m-1)\tau)], \quad (1)$$

with the measured voltage, $V(t_n)$, sampled at discrete $t_n = n\Delta t$ with $\Delta t = 12.5$ ps the experimental sampling period. In this study, we choose the embedding dimension $m = 3$ and the time-delay embedding $\tau \approx 1/(4f_0)$ which is close to the first zero of the autocorrelation function of $V(t)$[35].

**Poincaré section and symbolic analysis**. To simplify the definition of the Poincaré section ($x_3 = 0$) in the symbolic analysis, we apply a unitary transformation to the lag coordinates (which does not affect the topological equivalence between the reconstructed and original phase spaces) and form $\mathbf{x}_n^{(m)} = \mathbb{U}\mathbf{v}_n^{(m)}$, where $\mathbb{U}$ is a rotation matrix,

$$\mathbb{U} = \begin{bmatrix} 1 & 0 & 0 \\ 0 & \cos\theta_1 & -\sin\theta_1 \\ 0 & \sin\theta_1 & \cos\theta_1 \end{bmatrix} \begin{bmatrix} \cos\theta_3 & -\sin\theta_3 & 0 \\ \sin\theta_3 & \cos\theta_3 & 0 \\ 0 & 0 & 1 \end{bmatrix}, \quad (2)$$

with $\theta_1 = -20°$ and $\theta_3 = 67°$. The partition on the Poincaré section is set as $x_2 = 0.76x_1 - 0.005$ (Fig. 3a–c) delimiting two regions, $R_A$ and $R_B$. This choice, despite being arbitrary, allows us to capture both the incommensurate and commensurate regimes of the NCVO ($I_{dc} = 12.6 - 14.0$ mA). A reconstructed orbit is encoded with symbolic sequences, whose length is determined by the number of times the reconstructed attractor intersects the two regions.

**Computation of correlation dimension and entropy**. The correlation dimension $D_c$ provides insight on the fractal dimension of an attractor and hence is used for the detection of chaos from the filtered time series. Its computation relies on the Grassberger–Procacia (GP) algorithm[48] involving the correlation sum $C_m(\epsilon)$, which gives the average number of neighbouring vectors $\mathbf{x}_j^{(m)}$ within the range $\epsilon > 0$ from any given vectors $\mathbf{x}_i^{(m)}$ of the attractor obtained from the time-delay embedding procedure. It is defined as

$$C_m(\epsilon) = \frac{2}{(N - n_T)(N - 1 - n_T)} \sum_{i=1}^{N} \sum_{j=i+1+n_T}^{N} \Theta\left(\epsilon - \| \mathbf{x}_i^{(m)} - \mathbf{x}_j^{(m)} \|\right), \quad (3)$$

with $\epsilon$ representing the typical radius of the neighbourhood surrounding the vector $\mathbf{x}_i^{(m)}$, $\| \cdot \|$ the norm-2, and $\Theta$ the Heaviside function. To avoid the bias induced by finite-size effects of the time series and time-correlation of neighbouring vectors, we introduce the Theiler correction $|i - j| > n_T$ to select eligible neighbours for the computation[49]. The presence of self-similarity imposes the correlation sum to approximately satisfy a linear growth in log-scale and hence as a constant value (plateau) for its derivative. Hence, the correlation dimension $D_c$ is given by

$$D_c = \lim_{\epsilon \to 0} \lim_{N \to \infty} \frac{\partial \ln C_m(\epsilon)}{\partial \ln \epsilon}. \quad (4)$$

Using the correlation sum $C_m(\epsilon)$, it is also possible to compute a lower bound for the Kolmogorov–Sinai (KS) entropy $h_{KS}$ known as the $K_2$-entropy[50,51] and defined as

$$K_2 = \lim_{m \to \infty} \lim_{\epsilon \to 0} \lim_{N \to \infty} K_{2,m}(\epsilon) \text{ with } K_{2,m}(\epsilon) = \frac{1}{\tau} \ln\left[\frac{C_m(\epsilon)}{C_{m+1}(\epsilon)}\right], \quad (5)$$

with $\tau = n_\tau \Delta t$ the embedding time lag derived from the sampling time of the experimental time series[35]. The computation of $K_2$ can be also achieved by taking the limit of $K'_{2,m}(\epsilon) = 1/4\tau \ln\left[C_m(\epsilon)/C_{m+4}(\epsilon)\right]$, which is the quantity introduced in ref. [50] and used here. In practice, the approximate value of $K_2$ is determined by extrapolating the average value $\langle K'_{2,m}(\epsilon) \rangle_\epsilon$ on the scaling range (here $[1, 10^{0.3}]$) as $m$ takes larger values. In our analysis, we have used this approach after normalizing the experimental filtered time series as described in the previous section. We use $N = 1.2 \times 10^5$ samples and $n_T = 15$ for the Theiler correction, with the embedding dimension $m$ chosen between 10 and 24 for the correlation dimension and entropy calculations. The neighbourhood radii $\epsilon$ in the range $[10^{-1}, 10]$.

**Computation of the Shannon block entropy**. We consider a random source of $n$-bit words from the dictionary $\{s_i\}_{n_s}$ with $1 \leq n_s \leq 2^n$. The words are obtained by sliding a window of $n$ bits in width along the bit stream of length $N_b$ resulting from the symbolic analysis of the NCVO dynamics. The mathematical definition of the Shannon block entropy of the $n$-bit word source is given by

$$H_n = - \sum_{s_i \in \{s_1, \dots, s_{n_s}\}} p(s_i) \log_2 p(s_i), \quad (6)$$

with $p(s_i)$ the probability of appearance of symbol $s_i$. We can then determine the entropy per binary symbol (or entropy rate with a maximum value at 1 bit per symbol) using the limit $h = \lim_{n \to \infty} H_n / n$. In finite binary sequences of length $N_b$, the use of blocks of length $n$ usually leads to more robust estimates of the entropy compared with the direct estimation from the bit sequence. The probability of each word is determined with the likelihood estimator $\hat{p}(s_i) \approx \#(s_i)/N_s$ with $N_s = N_b - n + 1$. Finally, we use the upper limit for the block size given by

$$N_s h \geq n \, 2^n \ln 2 \quad (7)$$

for binary words as suggested in ref. [52]. Due to the finite size $N_b = 9300$ bits for the bit stream obtained from the NCVO's symbolic dynamics, we use block lengths in the range $n \in \{3, 6\}$.

**Computation of the Lempel–Ziv complexity**. The Lempel–Ziv complexity $C_{LZ}$ of a binary sequence measures the number of patterns present and is the basis of LZ77 compression[53,54]. For a large binary sequence of size $n$, it can be shown that the number of patterns $c(n)$ behaves asymptotically like the ratio $n/\log_2 n$. In a sequence of length $n \gg 1$, we can use this ratio as a normalization factor for the number of patterns in order to ensure that the complexity measure remains bounded in the range $C_{LZ} \in [0, 1]$. In order to compute $C_{LZ}$, we use the algorithmic procedure presented in ref. [55]. If the binary sequence is generated by a stationary and ergodic process, the Lempel–Ziv complexity coincides with the entropy rate $h$[52] in the limit of large $n$. Here, the complexity $C_{LZ}$ is computed from a sequence of $N_b = 9300$ bits.

**Reporting summary**. Further information on research design is available in the Nature Research Reporting Summary linked to this article.

## Data availability
The data sets generated and/or analysed during the current study are available from the corresponding author on reasonable request.

## Code availability
All relevant codes are available from the corresponding author on reasonable request.

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

## Acknowledgements
The authors thank Cyrile Deranlot and Stéphanie Girod for their assistance in film growth and sample preparation. This work was supported by the Agence Nationale de la Recherche under Contract No. ANR-17-CE24- 0008 (CHIPMuNCS), the Horizon2020 Research Framework Programme of the European Commission under Contract No. 751344 (CHAOSPIN), and the French RENATECH network. The Chaire Photonique is funded by the European Regional Development Fund (European Commission) (ERDF), Ministry of Higher Education and Research (FNADT), Moselle Department, Grand Est Region, Metz Metropole, AIRBUS-GDI Simulation, CentraleSupélec, and Fondation CentraleSupélec.

## Author contributions

J.-V.K., S.P.-W., V.C. and M.S. designed the study. K.B. and V.C. fabricated the samples. T.D. designed and implemented the experimental setup. M.-W.Y., J.L. and T.D. performed the high-frequency electrical measurements. M.-W.Y., D.R. and J.-V.K. analysed and interpreted the data. M.-W.Y. performed the simulations and interpreted the results. M.-W.Y. and D.R. prepared the manuscript. All authors edited and commented on the manuscript.

## Competing interests

The authors declare no competing interests.
