## [Peer Review File · Nature Communications]

Reviewers' Comments:

Reviewer #2:

Remarks to the Author:

Please see the supplemental PDF for the review.

Reviewer #3:

Remarks to the Author:

This is an interesting work demonstrating that deterministic chaotic behavior can be observed in the magnetization dynamics of a current-driven nanocontact existing in a vortex ground state.

The manuscript is original and contains sufficient new physics to justify its publication in the "Nature Communications"

My only critical comment is presented below:

It would be interesting to see a more detailed discussion on how the chaotic magnetization dynamics described in the manuscript could be used for information processing.

Authors' Rebuttal to Reviewer Reports – “Pattern generation and symbolic dynamics in a nanocontact vortex oscillator” (NCOMMS-19-27276)

M.-W. Yoo, D. Rontani, J. Létang *et al.*

November 21, 2019

In the following, we present a detailed response to each of the points raised by the Reviewers. For the sake of clarity and transparency, text from the Reviewer reports are drawn verbatim and are typeset in a gray box as shown below.

An example of text drawn from the Reviewer reports.

We respond to each point in the original order presented in the Reviewer reports.

We hope that our rebuttal and corresponding revisions will clarify all outstanding issues and offer a convincing argument for further consideration of our manuscript for publication in *Nature Communications*.

Reply to Reviewer 2

The reviewer writes:

The manuscript entitled “Pattern generation and symbolic dynamics in a nanocontact vortex oscillator” by Myoung-Woo Yoo et al. presents the results of an interesting set of experiments and analyses on nanocontact vortex oscillators (NCVOs). Through experiments and simulations, they show that for some parameter values, NCVOs can exhibit a peculiar (apparently chaotic) dynamical behavior: the system produces only a few relatively simple oscillatory patterns, but these patterns occur one after the other in an unpredictable sequence. The major claim is that these NCVOs can be tuned to exhibit maximal entropy (that is, to be perfectly unpredictable), and they suggest that this is a promising source of entropy, or randomness.

The idea that NCVOs can exhibit chaotic behavior is new but not unreported; indeed, the chaotic behavior of NCVO has been reported by many of the same authors in Ref. 27 just earlier this year. The idea that entropy can be harvested from chaotic physical systems is not new either; the authors accurately point this out as well (e.g., Refs. 2,4, and 5). The authors do perform a significantly more in-depth analysis of the dynamics observed in Ref. 27 than was reported there, resulting in a strong argument in favor of this being a system dominated by chaotic, rather than stochastic, behavior. In this manuscript, they emphasize the unpredictable pattern sequences, whereas they were not the focus of Ref. 27 and were only briefly mentioned there. Further, a system with this type of dynamical behavior may be particularly well-suited for harvesting entropy from the desired chaotic source (rather than from thermal fluctuations) because the entropy comes from the full pattern sequence rather than just the least significant bit of a single sample, as is often done. The authors may want to mention this point.

The chaotic behavior of unpredictable pattern sequences displayed by the NCVOs are fascinating, the agreement between the experimental and the simulated time series are impressive, and the time series analysis sufficiently convinces that the system is chaotic and well-characterizes the chaos. This is well- done in-depth study of a recently discovered chaotic system that may have interesting future applications. As the authors mention in the conclusion, this system is particularly well-suited for potential control of chaos experiments and investigations in the future, since the saddle is easily identified (as the end of each pattern) and so one could apply small perturbations there. It may also be well-suited for physical random number generation, although I have some doubts about this (see 1 below).

I think the manuscript is a significant step forward in understanding the dynamics of NCVOs. A major point the authors make, and a major question that I have about the NCVOs, is their practical utility as an entropy source for random number generation. I could recommend publication in Nature Communications once they have addressed this and the concerns below.

We thank the reviewer for their detailed appraisal of our work. We are encouraged by the overall positive remarks and we hope that our revisions will fully address the concerns raised, notably the practicality of NCVOs as physical sources of entropy.

Comments/Questions to the authors:

1. I have some concerns regarding the authors' claims that NCVOs are a promising source of entropy for information processing. Were they able to solidify this argument, I think the manuscript would have a much better chance of having a high impact.

a. The authors claim that NCVOs are highly compact, yet the experiments must be performed at 77 K. Cooling down to such temperatures typically requires a large apparatus, and it seems this would severely limit the practicality of such a device. Can the authors comment on this?

The Reviewer brings up a good point. We would like to stress that the device functions at room temperature, but the choice to conduct experiments at 77 K was made to improve the signal-to-noise ratio and to suppress as much as possible stochastic contributions to the oscillator signal. We have shown previously that the commensurate and incommensurate states can be observed under ambient conditions [Ref. 26 of the main text]. As such, we do not envisage any fundamental issues for exploiting the NCVO as an entropy source.

b. The authors claim that their random “bits are generated at an average rate of ~ 131 MHz, which is much faster than stochastic-based random number generators.” This statement is not correct. For example, PicoQuant sells a commercially available random bit generator (PQRNG 150) based on the time of arrivals of single photons (a stochastic process) that provides random bits at 150 Mbit/s rates, and amplified spontaneous emission has been shown to produce entropy at greater than 1 Gbit/s (e.g., see *Opt. Express* 18(23), 23584–23597 (2010) and *J. Lightwave Technol.* 30(9), 1329–1334 (2012)). Can the authors please clarify this claim?

We agree with the reviewer on this point. We were a little careless with our remarks, as we were focused on comparing the generation achieved by chaotic NCVOs with the rates from other stochastic, spintronic-based random number generators, which have much lower generation rate [Refs. 38 and 39 in the main text].

Indeed, when considering photonic- or electronic-based generator using stochastic effects, the generation rate can be much higher compared with our system. However, one of the advantages provided by spintronic devices is their sub 100 nm sizes and compatibility with conventional CMOS, thus allowing for large-scale, parallel implementation of this devices on a single chip, which could result in an aggregated bit rate comparable to state-of-the-art performance offered by other physical implementations. This strategy was previously applied using autonomous chaotic oscillators on an electronic microchip, where individual chaotic oscillators generated at most ~ 100 Mbit/s, but 128 of them were used in parallel for an aggregated generation rate of 12.8 Gbit/s [see, e.g., D. Rosin, D. Rontani, and D.J. Gauthier, *Ultrafast physical generation of random numbers using hybrid Boolean networks*, *Phys. Rev. E* **87**, 040902(R) (2013)]. A similar strategy could be envisaged with NCVOs, where their small form factor could allow for dense, parallel integration on a single chip, thus making possible aggregate generation rate from the tens up to the hundreds of Gbit/s.

In the revised manuscript, we have clarified our statement and detailed the point of comparison with spintronic-based devices and other technologies for physical random

generation.

c. While the fact that the entropy comes from a pattern sequence (rather than just the least significant bit of a single sample, as is often done) is attractive because you are assured that the entropy is indeed coming from the chaos (rather than from thermal from the NCVO (since the system has to oscillate many times to produce 1 bit of entropy). Can the authors address this? What are the limiting factors for increasing the entropy rate?

As the Reviewer mentions, the bits are produced by the chaotic polarity switching of the magnetic vortex core after several gyrations around the nanocontact, which is different with bit generation by temperature-induced stochastic dynamics at a single point. We have emphasize this point more clearly in the discussion section of the revised manuscript.

The limiting factor for the entropy rate is the core reversal rate, which is a fraction of the gyration frequency. This is inherent to the magnetization dynamics associated with topological solitons such as magnetic vortices. With our extraction technique, a single bit is produced after two core reversals, therefore, the bit-generation rate, f_{bit} is determined by the core-gyration frequency, f_0 , and required gyration number for the core switching, $f_{\text{bit}} = f_0 / (\langle n_+ \rangle + \langle n_- \rangle)$, where n_+ and n_- are required gyration numbers for the core switching in the cases of $p = +1$ and -1 , respectively, as shown in the Supplemental Note 1. Because $(\langle n_+ \rangle + \langle n_- \rangle)$ cannot be smaller than two, the generation rate would be mainly proportional to f_0 . Using our NCVO sample, we can reach the gyration frequency up to ~ 600 MHz, then the possible generation rate would be up to ~ 300 Mbit/s.

d. The pattern recognition algorithms seem to be computationally expensive. Can the authors comment on the computational complexity of their algorithm? Does this also limit the potential for a compact entropy source? The system may produce bits at ~ 100 MHz rates, but what's the rate at which you can extract the bits?

In the main text, we show the bit generation using the symbolic dynamics with filtered data which requires a lot of computation. As shown in the Supplemental Note 5, however, the bit can be extracted directly from non-filtered time traces by calculating convolutions using kernel functions (representative of the type of patterns generated by the NCVO) which are computed and stored in the memory of our broadband oscilloscope. The convolutions are then calculated using a simple software routine that could also be embedded in the oscilloscope or signal processing unit. The kernels are made of about 500 points in our case. In our algorithm, the convolution operation allows for the detection in the noisy signal of the type of pattern present, which is the closest to the kernel. In essence it acts in a similar way to a *matched filter*, a well known technique used in modern digital communications [see, e.g., J. G. Proakis and M. Salehi, *Digital Communications*, McGraw Hill, 5th Ed., (2007)]. Considering that the generation rate of the NCVO is in the 100 MHz range, it would be possible to implement this matched-filtering operation in real time using dedicated hardware platforms such as field programmable-gate arrays (FPGA) or digital signal processing boards to detect the pattern and generate the bits accordingly. While such efforts would indeed be an interesting follow up in terms of engineering chaotic NCVO as entropy sources in digital communication architectures, they remain well beyond the scope of our present study.

2. I have several questions regarding Figures 1f and 1g.

a. The y axes are labeled C_D , and the caption says that C_D is the correlation dimension; however, line 6 on page 5 says that the correlation dimension is represented by D_c . I think the C_D in the y-axis labels is supposed to refer to the correlation sum, which is denoted by $C(\epsilon, m)$ on page 5. Can the authors please clarify this?

We thank the referee for pointing out this typographical mistake. Indeed, in Figs. 1f and 1g, C_D refers to $C(\epsilon, m)$. D_c is the correlation dimension based on the definition of $C(\epsilon, m)$ to be computed using Eq. (4) in the method section. We have revised the manuscript to make this notation consistent and have chosen to replace $C(\epsilon, m)$ with $C_m(\epsilon)$ (this is a commonly used notation for the correlation sum in the literature).

b. In the same figures, there are many different lines for each of the values of I_{dc} . What do these different lines represent? If they are different embedding dimensions m , the authors should state so explicitly in the figure caption.

The referee is absolutely correct, the various lines represent the value of correlation sum $C_m(\epsilon)$ in Fig. 1f and $\partial \ln C_m(\epsilon) / \partial \ln \epsilon$ in Fig. 1g for increasing embedding dimensions $m = 6$ to $m = 10$. We have made corrected the caption regarding the figure (note also that we computed the correlation dimension for higher embedding dimension for a refined estimate of D_c in the revised manuscript, see answer to Question 2.d)

c. The argument of the Heaviside function in equation (3) should be $(\epsilon - |x_i - x_j|)$ (as in Ref. 35).

We thank the referee for pointing out this typographical mistake, indeed the argument of the Heaviside function should be indeed $(\epsilon - |x_i - x_j|)$. We have corrected this in the method section in the revised manuscript.

d. Is there a reason that the authors did not include $I_{dc} = 12.6$ mA in their correlation analysis, but did include it in all of their other analyses?

In Figure 1, we wanted to highlight the difference between the commensurate and incommensurate cases, and show that qualitatively different dynamics could be generated by the NCVO in these different regions. The correlation dimension analysis shows that the geometric dimension of the attractor is close to an integer value for the commensurate case (consistent) with a limit cycle dynamics and non-integer in the incommensurate case, consistent with the fractal geometry of a chaotic attractor. For $I_{dc} = 12.6$ mA, we expect a limit-cycle dynamics. The correlation dimension analysis will lead for correlation dimension curves superimposed with those from $I_{dc} = 14$ mA and impede the overall readability. For the sake of clarity, we retain only the analysis for $I_{dc} = 13.2$ mA and 14 mA, but we include the value of the correlation dimension at $I_{dc} = 12.6$ mA in the revised manuscript.

While conducting additional analyses at $I_{dc} = 12.6$ mA, we notice that by increasing the embedding dimension in the range $m = 10 - 24$ and focusing our analysis in the neighborhood radii in $[10^{-1}, 10]$ in the chaotic regime, the quantity $\partial \ln C_m(\epsilon) / \partial \ln \epsilon$ presented a larger ϵ -range with a plateau compared to our original analysis. This has

allowed us to average the near-constant value over the scaling range $[1, 10^{0.3}]$ and find a more precise estimation of the correlation dimension D_c . We have also given error bounds corresponding to 95% confidence intervals on all estimated values. We have reported these more precise estimates in the revised manuscript.

At $I_{dc} = 13.2$ mA, in the center of the incommensurate region, we find $D_c = 1.70 \pm 0.04$. This is still non-integer but is slightly less than the previous estimate found. We used similar numerical conditions for $I_{dc} = 12.6$ mA and 14 mA and have reported the corrected correlation dimension values: $D_c = 1.15 \pm 0.01$ and $D_c = 1.05 \pm 0.01$, respectively. Note that the value for $I_{dc} = 12.6$ mA is further away from integer value 1 because of the larger amount of jitter present in the time series compared to $I_{dc} = 14$ mA. Indeed, at lower DC currents, the core switching occurs after more gyrations, which may result in a larger jitter. This can be also seen when we compare the Poincaré sections at 12.6 mA and 14 mA in Figs. 3a and 3c, respectively; the Poincaré section at 12.6 mA presents a few points away from the main cycle and hence contributes to the detection of a small amount of fractal geometry by the correlation-based algorithm.

Note that we also clarified our notation of the logarithm in the revised manuscript and use the natural logarithm (\ln) instead of base-10 logarithm (NB: the choice of base affects the value of $C_m(\epsilon)$ but not $\partial \ln C_m(\epsilon) / \partial \ln \epsilon$, and hence the dimension D_c).

e. Have the authors performed correlation analyses for simulation data?

We thank the Reviewer for this suggestion. We did not originally include our additional analyses on the simulated time series. We calculated the correlation dimensions, D_c , for the simulation data in Supplementary Figure 2. The dimensions are almost 1 in the commensurate cases ($D_c = 1.05 \pm 0.01$ for $I_{dc} = 12.6$ mA and $D_c = 1.08 \pm 0.002$ for $I_{dc} = 13.3$ mA), while it has higher noninteger value in the incommensurate state ($D_c = 1.42 \pm 0.02$ for $I_{dc} = 13.1$ mA). This result is in good qualitative agreement with the experimental data. We state the result of the correlation dimension analysis in Supplementary Note 2.

3. Regarding the discussion of Fig. 4:

a. The authors state that “at $I_{dc} = 13.1$ mA, we have $P_{0,0} = P_{1,1}$ and $P_{0,1} = P_{1,0}$.” However, it looks to me that at $I_{dc} = 13.1$ mA $P_{0,0} = P_{0,1}$. At $I_{dc} = 13.0$ mA, I see that $P_{0,0} = P_{1,0}$ and $P_{0,1} = P_{1,1}$. Can the authors clarify what they mean?

We thank the Reviewer for their careful reading of our manuscript. There was indeed a mistake. Obviously, $P_{0,0} = P_{0,1} \approx 0.5$ and $P_{1,0} \approx 0.75$ and $P_{1,1} \approx 0.25$ at $I_{dc} \approx 13.2$ mA. We have corrected it in the revised manuscript. To avoid any misunderstanding, we have also removed the mention of ‘a fair coin toss’, although the NCVO can still be achieved a high entropy rate, > 0.9 bit/symbol, as shown in Fig. 4 and explained below.

A more in-depth analysis of entropy rate can support the above statement. According to information theory [see, e.g., T. M. Cover and J. A. Thomas, *Elements of Information Theory*, Wiley, 2nd Ed. (2006)], the entropy rate of a source $\{X_i\}_{i=1,n}$ is defined by

$$h_\infty = \lim_{n \rightarrow \infty} \frac{1}{n} H(X_1, \dots, X_n) = \lim_{n \rightarrow \infty} H(X_n | X_{n-1}, \dots, X_1). \quad (1)$$

For a memoryless source, the conditional entropy satisfies $H(X_n | X_{n-1}, \dots, X_1) =$

$H(X_n)$ and as a result the entropy rate is equal to the binary entropy,

$$h_\infty = -P_0 \log_2(P_0) - (1 - P_0) \log_2(1 - P_0), \quad (2)$$

which is maximum at 1 bit/symbol, if $P_0 = P_1 = 0.5$. Considering the Markovian property, we have $P(X_n|X_{n-1}, \dots, X_1) = P(X_n|X_{n-1})$, thus simplifying the mathematical definition of the entropy rate $h_\infty = \lim_{n \rightarrow \infty} H(X_n|X_{n-1})$, and as a result, we can exploit the transition probability in the computation of the entropy rate. The binary symbols generated by the NCVO can be described using a binary Markov-chain representation with the following transition matrix

$$\pi = \begin{pmatrix} P_{0,0} & P_{0,1} \\ P_{1,0} & P_{1,1} \end{pmatrix} = \begin{pmatrix} 1 - \alpha & \beta \\ \alpha & 1 - \beta \end{pmatrix}. \quad (3)$$

The stationary probability distribution is given here by $\mu : \left[\mu_0 = \frac{\alpha}{\alpha + \beta}; \mu_1 = \frac{\beta}{\alpha + \beta} \right]$. The entropy rate then reads here $h_\infty = -\sum_{i,j=0,1} \mu_i P_{i,j} \log_2 P_{i,j}$, which simplifies to

$$h_\infty = \frac{\beta}{\alpha + \beta} H_b(\alpha) + \frac{\alpha}{\alpha + \beta} H_b(\beta). \quad (4)$$

Where the Shannon block entropy and the LZ complexity are maximum (at $I_{dc} \approx 13.2$ mA), we have for the transition probabilities $\alpha = 0.4985$ and $\beta = 0.7430$. This leads to entropy rate for the NCVO:

$$h_\infty \approx 0.93 \text{ bit/symbol}. \quad (5)$$

This entropy-rate analysis of the NCVO is interesting on several levels. First, the entropy rate captures the asymptotic behavior of the bit extracted from the NCVO chaotic alternation of patterns and shows that despite the asymmetry in transition probability, very high levels of entropy per symbol (regardless of being "0" or a "1") can still be achieved (0.93 bit/symbols). Second, the level of entropy rate from the Markov Chain model is consistent with the level of entropy determined in our correlation analysis when we computed a lower bound for the KS entropy (see our response to point 4). Here, one symbol is generated approximately every 10 ns on average, so we end up having 0.093 bit/ns to be compared with 0.12 ns^{-1} for h_{KS} . Finally, the Markov chain representation is also interesting because it also allows us to interpret the limit in entropy generation per symbol from the transition between consecutive patterns, which highlights the impact of the difference in the physical properties of patterns leading to "1" and "0"; pattern "0" requires four gyrations around the nanocontact as opposed to three for pattern "1". As a consequence, the two patterns are not physically equivalent and we expect the transition $1 \rightarrow 0$ and $0 \rightarrow 1$, to be different from $0 \rightarrow 0$ and $1 \rightarrow 1$.

b. Can the authors comment on the type of bifurcations that are happening as the limit cycles apparently merge into a chaotic attractor, and essentially convert the system to a discrete-time system?

We thank the Reviewer for this very interesting question. We currently do not have a clear answer for the bifurcation mechanism leading to chaos. When observing Fig. 1b, for example, one can notice that between the commensurate region and the incommensurate region, there is a sudden spectral broadening, a signature of the chaotic regime. Because this transition is sudden, we might imagine a type of crisis bifurcation, but we do not have detailed analysis to characterize precisely the route to chaos (we are presently working toward developing a set of differential equations that capture the

observed dynamics to study this question in more detail). We can obtain more precise data from micromagnetic simulations at sufficiently close to the transition region; the results may provide information of the bifurcation mechanism. These are planned as future work to address this point.

4. The authors state that the NCVO inherits its complexity from the Kolmogorov-Sinai entropy created by the chaotic dynamics. Have they considered estimating the KS entropy (or at least the maximal Lyapunov exponent) from their simulations? Have they considered estimating the maximal Lyapunov exponent from experimental data?

We thank the Reviewer for this suggestion. Here, we propose to use the K_2 -entropy metric [P. Grassberger and I. Procaccia, Estimation of the Kolmogorov entropy from a chaotic signal, Phys. Rev. A **28**, 2591(R) (1983)] known to be a lower bound of the Kolmogorov-Sinai (KS) entropy [J.-P. Eckmann and D. Ruelle, Ergodic theory of chaos and strange attractors, Rev. Mod. Phys. **57**, 617 (1985)] and we compute it on the experimental and simulation data. Like the correlation dimension, the K_2 -entropy can also be computed from the correlation sum and satisfies the following relationship

$$K_2 = \lim_{N \rightarrow \infty} \lim_{\epsilon \rightarrow 0} \lim_{m \rightarrow \infty} K_{2,m}(\epsilon) \text{ with } K_{2,m}(\epsilon) = \frac{1}{\Delta t} \ln \left[\frac{C_m(\epsilon)}{C_{m+1}(\epsilon)} \right], \quad (6)$$

where Δt is time-series sampling time, N the number of samples in the time series, ϵ the length of neighboring radii, and $C_m(\epsilon)$ the correlation sum evaluated for embedding dimension m . Note that if the embedding time-delay $\tau = n_\tau \Delta t$ is larger than the sampling time ($n_\tau > 1$), then $K_{2,m}(\epsilon)$ needs to be rescaled with respect to τ as indicated in T. Schreiber and H. Kantz [*Nonlinear time series analysis*, Cambridge University Press, 2nd Ed. (2004)].

For large values of the embedding dimension, we search for approximately constant values in the scaling range $\epsilon \in [1, 10^{0.3}]$ and take the average $\langle K_{2,m}(\epsilon) \rangle$, where the fractal nature of the chaotic attractor of NCVO shows up in the correlation sum. Finally, we look at the asymptotic behavior $\langle K_{2,m}(\epsilon) \rangle$ with increasing embedding dimension m to get an approximate estimation of K_2 (with limited amount of data points, we need to extrapolate for larger values of m). For this, we use an exponential fit of the form $a_1 + a_2 e^{-a_3 m}$ and determine the coefficients a_i by nonlinear regression.

Another quantity other than $K_{2,m}(\epsilon)$ can be used to conduct a similar estimation and has been considered and used in Grassberger and Procaccia for K_2 -entropy estimation; this is

$$K'_{2,m}(\epsilon) = \frac{1}{4\Delta t} \ln \left[\frac{C_m(\epsilon)}{C_{m+4}(\epsilon)} \right]. \quad (7)$$

We have studied the evolution of the metric $K'_{2,m}(\epsilon)$ for embedding dimension $m = 10$ to 24. By averaging the value of $K'_{2,m}(\epsilon)$ in the ϵ -range $[1, 10^{0.3}]$ we observe an asymptotic evolution of the quantity as m increases. Similar to what is done in Grassberger and Procaccia, we extrapolate to infer the value of K_2 as m becomes large and we find that $K_2 \sim 0.12 \text{ ns}^{-1}$.

This value of the K_2 -entropy gives a lower bound for the KS entropy (i.e., average loss of information in the chaotic NCVO). This will guarantee that at approximately 100 MHz, we do not extract more entropy than what the chaotic NCVO can generate. This provides further evidence that the bit/pattern extraction technique employed here

allows us to capture the entropy generated from the chaotic polarity switching of the magnetic vortex core generated by the NCVO. We have included this information in the revised manuscript, added two additional sub-figures in Figure 1, and modified the method section to account for the calculation of the K_2 -entropy.

5. The authors use Shannon Block Entropy and Lempel-Ziv complexity to argue that their NCVO produces maximal entropy for certain parameters. Typically NIST standards (often the 2012 standards, but now the new SP 800-90B Recommendation for the Entropy Sources Used for Random Bit Generation may be more appropriate) are used to argue for randomness. Have the authors considered testing their bit sequences with such a test suite?

We thank the Reviewer for the suggestion. Indeed, testing the randomness using the NIST statistical test suites would allow the suitability of the chaotic bit generated by the core-polarity switching events as random numbers to be assessed. Due to experimental constraints, the largest one-shot time series acquisition is limited to approximately 10 kbits at the moment, whereas the statistical test suite SP800-90B recommends at least 1 Mbits, and for the more standard NIST test suite NIST SP800-22, 1 Gbits is usually recommended. Hence, any analysis based on these suites must be treated with caution because we have limited statistics.

Nevertheless, using the data available (3097 bits extracted from experimental data at $I_{dc} = 13.2$ mA, after applying von Neumann debiasing and XOR post-processing), we have performed the tests from the SP800-90B suite as suggested by the Reviewer. We have compared these results with a software-based pseudorandom number generator (Mersenne twister implemented in *Mathematica 12.0*) with the same bitstream length, in order to facilitate a comparison with limited statistics. The results are reported in Table 1.

Method	Min-entropy NCVO (debaised)	Min-entropy Mathematica
Most Common Value Estimate	0.917458	0.913945
Collision Estimate	0.569646	0.568525
Markov Estimate	0.930367	0.948814
t-Tuple Estimate	0.836739	0.840302
LRS Estimate	0.916016	0.933154
Lag Prediction Estimate	0.920524	0.939163
MultiMMC Prediction Estimate	0.908655	0.914789
LZ78Y Prediction Estimate	0.890487	0.919401

Table 1: Results of the NIST SP800-90B test suite on the debaised NCVO bits generated in the incommensurate state versus the `RandomInteger[]` function in *Mathematica 12.0* using the Mersenne Twister. Test realized with 3097 bits.

One can see that under similar conditions, the software-based generator performs marginally better on most tests, but the overall performance is comparable to that of the NCVO. Similarly to what is done elsewhere in electronics and photonics, where high-speed random number generator have been demonstrated, one would have to analyze in depth the proper post-processing (e.g. multi-stage XOR, LFSR etc) techniques necessary to remove inherent biases, unbalanced proportions of ‘0’ and ‘1’, or residues of patterns in the NCVO bit sequences. This is a very interesting application of our theoretical analysis, but beyond the scope of this article.

Because we do not have sufficient statistics to satisfy the prerequisites of the NIST SP800-90B tests, we prefer not to include this analysis in our manuscript or supplementary material.

Minor comments:

p. 7 line 19: “However finding the proper (said to be generating) from model-free...” A word is missing after “proper.”

We have made the correction accordingly in the revised manuscript.

How did the authors choose the time-delay embedding tau?

We chose $\tau = 1/(4f_0)$ as explained in Methods section, because the measured time series, $V(t)$, is a projection of the vortex core motion, so its orthogonal component should have a 90° phase difference. Note that the first zero crossing of the autocorrelation is often chosen as a lag time τ , and it is also near $\tau = 1/(4f_0)$.

The authors should include the relevant reference for the Theiler condition for the correlation sum calculation.

We thank the referee for pointing this out. We have included the following reference for the Theiler correction (which is one of the seminal papers discussing this approach):

- J. Theiler, Spurious dimensions from correlation algorithms applied to limited time-series data, *Phys. Rev. A* **34**, 2427 (1986).

Reply to Reviewer 3

The Reviewer writes:

This is an interesting work demonstrating that deterministic chaotic behavior can be observed in the magnetization dynamics of a current-driven nanocontact existing in a vortex ground state.

The manuscript is original and contains sufficient new physics to justify its publication in the “Nature Communication.”

My only critical comment is presented below:

It would be interesting to see a more detailed discussion on how the chaotic magnetization dynamics described in the manuscript could be used for information processing.

We thank the referee for the positive remarks and recommendation for publication.

The chaotic information generation using the NCVO can be applicable to various areas as mentioned in the introduction and conclusion of the main text, such as encrypted communications, encoding information with symbolic dynamics, random number generation, and data processing. Applications of chaos are well established in electronics and in photonics. However, the chaotic dynamics of magnetization has specific features that makes it particularly interesting for specific classes of applications.

The first important feature is that the complexity of the magnetization dynamics is not encoded in the amplitude of the waveform but in the alternation of regular patterns. In that sense, it natively exhibits very similar properties to the chaos generated by specifically-engineered systems proposed by Corron, Blakely, and Stahl [A matched filter for chaos, *Chaos* **20**, 023123 (2010)] and hence has high resilience to perturbations. The extraction of the bits, once the two patterns are identified and stored digitally, could be done in real-time with a low computational cost using matched-filters (see, e.g., J. G. Proakis & M. Salehi, *Digital Communications* (McGraw-Hill, 2007), 5th ed.) implemented on field programmable-gate arrays (FPGA) or digital signal processors (DSP). A recent application of chaotic dynamics similar to those of NCVOs was proposed for WIFI communications (i.e., for challenging environments with multi-path interference, jamming, distortion) using electronic circuits [see, e.g., H. Ren, M. Baptista & C. Grebogi, *Wireless communication with chaos*, *Phys. Rev. Lett.* **110**, 184101 (2013); H. Ren, C. Bai, J. Liu, M. Baptista, & C. Grebogi, *Experimental validation of wireless communication with chaos*, *Chaos* **26**, 083117 (2016)], because of the possibility to recover more easily regular patterns in low signal-to-noise conditions compared with chaotic features in the amplitude and/or phase of a signal.

A second feature is that the NCVO generates directly chaotic bits with high levels of entropy (e.g., entropy rate $h_\infty = 0.93$ bit/binary symbol at $I_{dc} = 13.2$ mA in the incommensurate state), hence making it a promising avenue for physical random number generation with minimal post-processing. Another advantage is that chaotic effects take place at the nanoscale, hence allowing for dense integration of NCVOs on a single spintronic chip. This could allow for the parallel generation of random bits with aggregate rates in the tens or hundreds of Gbit/s (although each NCVO can alone generate at most 100 Mbit/s) similar to the approach already used with 128 parallel random generator implemented in parallel on a single microelectronic circuits [D. Rosin,

D. Rontani & D. J. Gauthier, Ultrafast physical generation of random numbers using hybrid Boolean networks, *Phys. Rev. E* **87**, 040902 (2013)].

Finally, the properties of the chaotic magnetization dynamics of NCVOs can be easily tailored with the DC current injected in the nanocontact. As a result, one could design advanced chaos control strategies by injecting small perturbations in the current to control the chaos and encode data in pattern alternation as described by Hayes, Grebogo, and Ott [Experimental control of chaos for communication, *Phys. Rev. Lett.* **73**, 1781 (1994)] to create a robust chaos-based encryption at the physical layer.

These elements of discussion and the associated references have been used as a basis for an additional paragraph (just before the conclusion), where the specific features of the chaotic dynamics of magnetization are detailed for application in information processing.

REVIEWERS' COMMENTS:

Reviewer #2 (Remarks to the Author):

In my opinion, the revised manuscript addresses the questions and comments brought forth by the reviewers. I recommend the this very nice work for publication in Nature Communications.

One very minor note: The dotted lines in Fig. 1g are quite difficult to see. It might be worth making them a little thicker and/or pointing them out with arrows on the side of the figure. However, the aesthetics of course are for the authors to decide.

Reviewer #3 (Remarks to the Author):

In the opinion of this Referee, the authors in the course of revision addressed most of the comments of both original Referees, and the revised version of their manuscript can be published in the Nature Communications as is.

Author's response to Reviewer Reports

Reply to Reviewer #2

The Reviewer writes:

"In my opinion, the revised manuscript addresses the questions and comments brought forth by the reviewers. I recommend the this very nice work for publication in Nature Communications.

One very minor note: The dotted lines in Fig. 1g are quite difficult to see. It might be worth making them a little thicker and/or pointing them out with arrows on the side of the figure. However, the aesthetics of course are for the authors to decide."

We thank the Reviewer for this positive appraisal and are delighted that they recommend publication in Nature Communications. We have followed the Reviewer's suggestion and modified Fig. 1g accordingly.

Reply to Reviewer #3

The Reviewer writes:

"In the opinion of this Referee, the authors in the course of revision addressed most of the comments of both original Referees, and the revised version of their manuscript can be published in the Nature Communications as is."

We thank the Reviewer for their evaluation and are thrilled with their recommendation to publish as-is.